# Significantly Reduced Retinol Binding Protein 4 (RBP4) Levels in Critically Ill COVID-19 Patients

**DOI:** 10.3390/nu14102007

**Published:** 2022-05-10

**Authors:** Richard Vollenberg, Phil-Robin Tepasse, Manfred Fobker, Anna Hüsing-Kabar

**Affiliations:** 1Department of Medicine B for Gastroenterology, Hepatology, Endocrinology and Clinical Infectiology, University Hospital Muenster, Albert-Schweitzer-Campus 1, 48149 Muenster, Germany; phil-robin.tepasse@ukmuenster.de (P.-R.T.); anna.huesing-kabar@ukmuenster.de (A.H.-K.); 2Center for Laboratory Medicine, University Hospital Muenster, 48149 Muenster, Germany; manfred.fobker@ukmuenster.de

**Keywords:** COVID-19, retinol binding protein 4, vitamin A, retinol, retinoic acid, ARDS, pneumonia, pandemic, SARS-CoV-2, inflammation

## Abstract

The SARS-CoV-2 virus is the causative agent of the COVID-19 pandemic. The disease causes respiratory failure in some individuals accompanied by marked hyperinflammation. Vitamin A (syn. retinol) can exist in the body in the storage form as retinyl ester, or in the transcriptionally active form as retinoic acid. The main function of retinol binding protein 4 (RBP4), synthesized in the liver, is to transport hydrophobic vitamin A to various tissues. Vitamin A has an important role in the innate and acquired immune system. In particular, it is involved in the repair of lung tissue after infections. In viral respiratory diseases such as influenza pneumonia, vitamin A supplementation has been shown to reduce mortality in animal models. In critically ill COVID-19 patients, a significant decrease in plasma vitamin A levels and an association with increased mortality have been observed. However, there is no evidence on RBP4 in relation to COVID-19. This prospective, multicenter, observational, cross-sectional study examined RBP4 (enzyme-linked immunosorbent assay) and vitamin A plasma levels (high-performance liquid chromatography) in COVID-19 patients, including 59 hospitalized patients. Of these, 19 developed critical illness (ARDS/ECMO), 20 developed severe illness (oxygenation disorder), and 20 developed moderate illness (no oxygenation disorder). Twenty age-matched convalescent patients following SARS-CoV-2 infection, were used as a control group. Reduced RBP4 plasma levels significantly correlated with impaired liver function and elevated inflammatory markers (CRP, lymphocytopenia). RBP4 levels were decreased in hospitalized patients with critical illness compared to nonpatients (*p* < 0.01). In comparison, significantly lower vitamin A levels were detected in hospitalized patients regardless of disease severity. Overall, we conclude that RBP4 plasma levels are significantly reduced in critically ill COVID-19 patients during acute inflammation, and vitamin A levels are significantly reduced in patients with moderate/severe/critical illness during the acute phase of illness.

## 1. Introduction

Severe acute respiratory syndrome coronavirus-2 (SARS-CoV-2) is the pathogen of the coronavirus disease (COVID-19) pandemic [1]. While the disease course is mild in most cases, some patients develop a severe disease course accompanied by acute respiratory distress syndrome (ARDS), severe lung injury, multiple organ failure, and high mortality [2,3,4]. Histologically, diffuse alveolar damage was demonstrated in ARDS-COVID-19 patients. Here, features of the exudative and proliferative phase with interstitial and intra-alveolar edema, dying pneumocytes, hyaline membranes, microvascular thrombosis, capillary congestion, and AT2 cell hyperplasia could be detected [5]. In some patients, a massive hyperinflammation is induced with similarities to classic cytokine storm syndromes [6].

Vitamin A (syn. retinol) can be esterified in the body to the vitamin A storage form retinyl esters or oxidized to retinoic acid. The latter is a transcriptionally active form that is a ligand for certain nuclear hormone receptors, the retinoic acid receptors (RAR). Retinol, retinyl esters, and retinoic acid are collectively referred to as retinoids [7,8]. Vitamin A plays a central role in the innate and adaptive immune response [9]. The vitamin is essential for embryonic lung development and lung-tissue repair after infection-related injury [10,11]. It may also play an important part in the cure from COVID-19 pneumonia. The cellular infection mechanism of the SARS-CoV-2 virus is mediated by the cell surface receptor angiotensin-converting enzyme 2 (ACE2) [12,13,14]. ACE2-mediated cell entry is followed by serine protease-transmembrane serine protease 2 (TMPRSS2)-dependent S-protein priming [15]. In the setting of acute infection, an interferon-induced increase in ACE2 expression is thought to occur in type 2 alveolar cells and germinated cells [16]. This may promote further viral spread [16]. The regulation of the ACE2 receptor in the context of SARS-CoV-2 infection could result from the dysregulation of the ACE2-angiotension II balance [17,18]. This dysregulation potentially has a negative effect on the function of the innate immune system [14]. The vitamin A metabolite all-trans-retinoic acid upregulates the expression of ACE2 and consequently interferes with the ACE2-angiotension II balance [19,20]. On the one hand, this could increase the risk of viral infection at the time of viral exposure [21], and on the other hand, it could reduce ACE2-Ang-1-7-Mas-dependent sympathetic overactivation [22,23,24]. In particular, the latter is observed in critical disease courses of obese or diabetic patients [14]. Vitamin A can be found in various forms in the human body. Approximately 80% is stored in the hepatic stellate cells [25]. Vitamin A binds to lipoproteins [26], albumin [27], and the transport proteins retinol-binding proteins 1–4 (RBP1–4) [28]. RBP1 functions as an intracellular regulator of vitamin A metabolism, serves in retinoid transport, and is essential for visual function [29]. RBP2 is mainly localized in the absorptive cells of the proximal small intestine and regulates retinoid uptake from food [30]. The transport of vitamin A from photoreceptors to the retinal pigment epithelium is RBP3-mediated. It is essential for the normal function of rods and cones. [31]. However, vitamin A circulates largely as retinol bound to RBP4, which is mainly synthesized by hepatocytes. After transport to tissues, it is released and transformed into retinoic acid. This provides a ligand for multiple nuclear receptors [32,33]. Since the transport of retinol by RBP4 is the major pathway for its distribution, RBP4 can be used as a marker for tissue delivery of vitamin A [7].

Studies investigating vitamin A plasma levels and RBP as surrogate markers in COVID-19 patients in relation to disease severity are currently not available. Therefore, the aim of this study was to characterize vitamin A plasma levels and RBP as surrogate parameters in acute COVID-19 and to analyze the relationship between plasma levels and disease severity and progression.

## 2. Materials and Methods

### 2.1. Study Participant Selection and Patient Samples

In this prospective study with a multicenter cross-sectional design, 283 hospitalized COVID-19 patients (detection of SARS-CoV-2 infection in nasopharyngeal swab, test by polymerase chain reaction) from Münster University Hospital and Marien-Hospital Steinfurt in Germany were included (03/2020–02/2021). Blood was collected from hospitalized patients at the time of the acute phase of illness. The laboratory data and medical history were taken from the patient records. Of the 283 patients, 59 were manually selected by matching for age and gender between groups. COVID-19 disease severity was defined using the World Health Organization (WHO) severity categories: critical (need for life-sustaining treatment, evidence of acute respiratory distress syndrome (ARDS), sepsis, septic shock; *n* = 19), severe (evidence of oxygenation failure, signs of pneumonia, respiratory distress; *n* = 20), or moderate (absence of signs of severe/critical disease progression; *n* = 20). ARDS was diagnosed as described by the Berlin definition [34]. Blood sampling was done during the initial 24 h after hospitalization. In addition, 91 subjects were included in the study after mild SARS-CoV-2 infection without hospitalization. Blood sampling was performed in convalescents. Of these, 20 subjects were manually selected for age and sex matching. Plasma samples were collected after written informed consent was obtained. The study was performed in accordance with the Declaration of Helsinki of 1975 as amended in 1983 and approved by the Ethics Committee of the University of Münster (AZ 2020-220-f-S, AZ 2020-210-f-S). Plasma samples were shielded from light and stored frozen (−80 °C) until measurement.

### 2.2. Laboratory Measurements

Serum retinol-binding protein 4 (RBP4) was measured by enzyme-linked immunosorbent assay (Immundiagnostik AG, Bensheim, Germany). According to the manufacturer, the reference ranges for adults are 20–75 mg/L. Vitamin A was determined on EDTA plasma using a commercial high-performance liquid chromatography kit (Chromsystems, Munich, Germany) according to the manufacturer’s directions. Vitamin A levels were given in mg/L. The blood sample was not taken while fasting. Laboratory tests measured complete blood count and levels of ferritin, interleukin-6, procalcitonin, gamma-GT, creatinine, C-reactive protein (CRP), lactate dehydrogenase (LDH), albumin, alanine aminotransferase (ALT), and pseudocholinesterase (PCHe).

### 2.3. Data Analysis/Statistics

For categorical variables, Fisher’s exact test or chi-square test was performed (presented as absolute numbers, percentages). Continuous variables were compared using the Mann–Whitney U test (Wilcoxon test, showing the median and the interquartile range). A Kruskal–Wallis test was conducted to compare more than two groups. Subgroup analysis was performed by a Bonferroni correction post hoc test (Levene test, for equal variance) or Games–Howell test (unequal variance). The Pearson correlation coefficient was used to correlate RBP4 and vitamin A levels with other parameters. The tests performed were two-tailed tests, and a *p* < 0.05 was interpreted as an indication of a statistically significant difference. Statistical analyses were conducted using SPSS 26 (IBM, Armonk, NY, USA).

## 3. Results

### 3.1. Cohort Characteristics

The hospitalized and convalescent COVID-19 patients had a median age of 58 years (min 21 years, max 88 years), 83% of the study participants were male. There were no significant differences (*p* > 0.05) between the study groups (hospitalized patients with moderate, severe, and critical illness and convalescent patients) with respect to these characteristics. The BMI (kg/m^2^) of patients with critical disease course and in convalescent patients was higher than in patients with a moderate or severe course (*p* < 0.001).

The interval between symptom onset or SARS-CoV-2 first detection and blood collection was 9 (3–16) days in hospitalized patients and 46 (42–53) days in convalescent patients (*p* < 0.001). Patients with blood collection in convalescence after a mild COVID-19 disease progression were not previously ill and were not taking regular medication. There were no significant differences between hospitalized patient groups with respect to pre-existing cardiovascular disease, respiratory disease, kidney insufficiency, neoplasm, and diabetes mellitus (*p* > 0.05). While 32% of the critically ill COVID-19 patients died during the hospital stay, no deaths occurred in the other patient groups (*p* < 0.001). Patients with blood sampling in convalescence had median normal inflammation, liver, and renal retention parameters. Depending on disease severity, patients with critical, severe, or moderate disease had median severity-dependent elevations of inflammation, liver, and renal retention parameters (Table 1).

### 3.2. COVID-19 Patients: Retinol-Binding Protein 4 and Vitamin A Plasma Levels

The comparison of all hospitalized COVID-19 patients (moderate, severe, critical course) at the time of acute disease phase with convalescent patients showed significantly decreased RBP4 plasma levels (hospitalized patients: 17.02 mg/L (11.5–22.6 mg/L) vs. convalescent patients: 21.8 mg/L (19.6–23.8 mg/L), *p* < 0.01) and vitamin A levels (hospitalized patients: 0.34 mg/L (0.18–0.42 mg/L) vs. convalescent patients: 0.65 mg/L (0.5–0.77 mg/L), *p* < 0.001; Figure 1). There were no significant differences between males and females with respect to RBP4 and vitamin A levels. In hospitalized patients, blood sampling occurred at a median of 4–13 days after symptom onset; in convalescent patients, blood sampling occurred at a median of 46 days after symptom onset (*p* = 0.003).

In the subgroup analysis of hospitalized patients with a critical disease course (14.6 mg/mL (10.0–21.7 mg/mL)), there were significantly lower RBP4 plasma levels compared to convalescent patients (21.80 mg/mL (19.56–23.84 mg/mL); *p* < 0.05). There were no significant differences between hospitalized patients with moderate (19.38 mg/mL (14.59–25.28 mg/mL)) or severe course (15.13 mg/mL (10.23–20.39 mg/mL)) compared to convalescent patients (21.80 mg/mL (19.56–23.84 mg/mL); *p* > 0.05)) regarding RBP4 plasma levels. A subgroup analysis of hospitalized COVID-19 patient groups showed no significant differences of RBP4 plasma levels between groups (moderate, severe, and critical, *p* = 0.218) (Figure 2a).

In comparison with convalescent patients, hospitalized COVID-19 patients with critical (0.26 mg/mL (0.17–0.33 mg/mL)), severe (0.37 mg/mL (0.19–0.53 mg/mL)), and moderate disease progression (0.37 mg/mL (0.26–0.45 mg/mL)) showed significantly lower vitamin A levels (0.65 mg/mL (0.51–0.77 mg/mL), *p* < 0.001; Figure 2b). Vitamin A levels did not differ significantly between hospitalized patient groups depending on disease severity (*p* > 0.05; Figure 2b). Considering the molar RBP4/vitamin A ratio, hospitalized patients showed significantly higher levels during the acute phase of illness (Figure 2c). Of the hospitalized COVID-19 patients, critically ill patients had significantly higher levels compared with severely ill patients (*p* < 0.01, Figure 2c).

### 3.3. Correlation of Retinol-Binding Protein 4 Plasma Levels with Laboratory Parameters

In the overall study group (*n* = 79), reduced retinol-binding protein 4 plasma levels correlated significantly with reduced vitamin A plasma levels (r = 0.68, *p* < 0.001. Reduced RBP4 levels correlated significantly with increased CRP levels (r = −0.322, *p* = 0.004), lymphocytopenia (r = 0.250, *p* = 0.025) Elevated creatinine (r = 0.426, *p* < 0.001) levels also significantly correlated with low RBP4 levels. Increased liver transaminases (ALT, r = 0.221, *p* = 0.048) and lower albumin levels (r = 0.250, *p* = 0.026) also significantly correlated with low RBP4 levels (Figure 3a–f). Low vitamin A levels also significantly correlated with increased CRP levels (r = −0.435, *p* < 0.001) and increased ferritin levels (r = −0.315, *p* = 0.004). Transaminases (AST r = −0.343, *p* = 0.002), y-GT (r = −0.222, *p* = 0.048), PCHe (r = 0.547, *p* < 0.001), albumin (r = 0.614, *p* < 0.001) also significantly correlated with vitamin A.

## 4. Discussion

This study investigated the connection between RBP4 and vitamin A plasma levels with inflammatory markers and the progression of COVID-19 (critical, severe, moderate) in the acute phase of the disease in hospitalized patients and in convalescent patients.

The inflammatory markers studied (leukocyte counts, lymphocyte counts, ferritin, interleukin-6, CRP) correlated significantly with disease severity in the hospitalized patients. Liver function (PCHe, albumin) was found to be significantly impaired depending on the course of the disease. These data confirm previous studies demonstrating liver injury in SARS-CoV-2 infections [35].

In our study, we demonstrated significantly reduced RBP4 (*p* < 0.01) and vitamin A levels (*p* < 0.001) in the group of hospitalized COVID-19 patients in the acute phase of COVID-19 infection compared to convalescent patients after a mild course. In a subgroup analysis, we demonstrated significantly reduced RBP4 levels in critically ill COVID-19 patients compared to convalescent patients. In contrast, RBP4 levels did not differ significantly in patients with a severe or moderate disease. In comparison, vitamin A levels were significantly decreased in hospitalized patients with critical (*p* < 0.001), severe (*p* < 0.001), and moderate (*p* < 0.001) disease progression compared with convalescent patients. Considering the entire study cohort, we demonstrated a significant, tight correlation of decreased RBP4 and vitamin A levels (*p* < 0.001). The RBP4/vitamin A molar ratio was increased in hospitalized COVID-19 patients during the acute phase of illness. This suggests that more apo-RBP4 circulates in plasma in hospitalized COVID-19 patients [36].

The liver has a key function in vitamin A metabolism. Approximately 80% of all retinoids in the human body are stored there [37], and the majority of RBP4 is expressed there [25]. A severe COVID-19 disease can lead to damage in multiple organs. In addition to cardiac injury and renal damage, liver damage has also been described. The mechanisms of association between liver injury and SARS-CoV-2 infections are complex and the subject of current research. Direct cholangiocyte injury has been described in the context of cytokine growth triggered by SARS-CoV-2 [38]. We also confirmed these data in our study, where critically ill COVID-19 patients showed significantly increased liver enzymes. In addition, elevated liver enzymes (ALT) and reduced albumin levels, a sign of liver synthesis efficiency, correlated significantly with reduced RBP4 levels. A possible explanation of the reduced RBP4 levels in critically ill COVID-19 patients could be SARS-CoV-2-related damage or an impairment of liver function. The association of reduced hepatic vitamin A mobilization during acute inflammation in critically ill COVID-19 patients remains speculative but is conceivable and requires further investigation.

Most of the vitamin A is bound to the protein RBP4 [39]. Considering the entire study cohort, we also demonstrated a significant tight correlation of decreased RBP4 and vitamin A levels (*p* < 0.001). This highlights the importance of RBP4 as a surrogate parameter for assessing vitamin A balance [40,41]. These results confirm existing data demonstrating decreased vitamin A levels in the setting of severe infection and inflammation due to increased urinary vitamin A loss [12], decreased mobilization of the vitamin in the liver [42], and decreased intestinal absorption [43]. In our study, we demonstrated a significant correlation of elevated inflammatory markers such as CRP elevation (*p* = 0.02) and lymphocytopenia (*p* = 0.04) with reduced plasma RBP4 levels. Both are established markers of disease activity and predictors of worse outcome in COVID-19 patients [44]. In addition, the recognition mechanism of SARS-CoV-2 RNA by retinoic-acid-inducible gene-1 receptors (RGI-1) may lead to the consumption of the retinoid reserve [45]. The immunopathogenesis of COVID-19 is characteristic of acute respiratory distress syndrome and multi-organ involvement with impaired type I interferon response and hyperinflammation. Retinoid insufficiency is thought to lead to impaired retinoid signaling and the interruption of interferon-1 synthesis. This may cause an exaggerated inflammatory response. Furthermore, the binding of vitamin A to the fatty acid binding site in the SARS-CoV-2 spike protein may possibly stabilize the occluded spike conformation. This could reduce the possibility of an ACE-2 interaction. Consequently, the restoration of retinoid signaling and vitamin A binding to the fatty acid binding site in the SARS-CoV-2 spike protein may represent another strategy for the treatment of COVID-19 [46].

Vitamin A, in its function as a T-cell effector, plays an important role in innate and adaptive immunity and stimulates the expression of interferon-stimulated genes (e.g., retinoic-acid-inducible gene 1, IFN-regulatory factor 1 (IFN-1)) [47]. Previous studies have shown a protective effect of retinoids on the progression of viral diseases including hepatitis B (HBV), norovirus, MeV, influenza, and cytomegalovirus [47]. In patients with influenza virus infection, a positive correlation of retinol-binding protein (RBP) as a vitamin A surrogate parameter and influenza-specific antibody neutralization titers has been demonstrated [48]. The Am580-mediated disruption of sterol-regulatory-element-binding-protein (SREBP)-dependent lipogenic pathways has been demonstrated in Middle East respiratory syndrome coronavirus (MERS-CoV) and SARS-CoV infections. Both viruses appear to stop IFN-1 mediated antiviral recruitment [49,50,51,52]. Retinoid use has been shown to increase IFN-1 as a potent antiviral cytokine [8]. Several studies have shown that vitamin A supplementation can reduce the risk of severe illness and death, as demonstrated in children with measles and in influenza pneumonia (mouse model) [11]. Severe infections and inflammation correlate with reduced vitamin A levels due to increased urinary vitamin A loss [12], decreased mobilization of the vitamin in the liver [42], and decreased intestinal absorption [43]. 

There are some limitations to this study. In our study, the comparison group of convalescent COVID-19 patients had a higher median BMI. This could be a limitation of the study. In addition to the liver, adipose tissue has a function in vitamin A metabolism and storage. In animal models, RBP4 synthesis and secretion have been demonstrated in adipocytes [53]. Whether RBP4 is synthesized primarily in visceral or subcutaneous adipose tissue is currently the subject of controversial research [54]. Whether BMI has an effect on RBP4 synthesis is sometimes unclear and controversial. In the study by Baljzová et al. (2008) [55], no significant differences were demonstrated in a small cohort with respect to RBP4 levels in obese and nonobese women [56,57,58,59,60,61]. Only bound retinol was detected with the method used; esters could not be detected. Since the patients were not fasting at blood collection, this is an important limitation. Our paper provides only descriptive data that do not demonstrate causality. Further studies are needed to clarify the presumed association between decreased serum vitamin A levels and critical COVID-19 disease progression. In addition, the sample size was small, especially in the cohort subgroups, which may introduce bias. Given these limitations, the consequences of reduced RBP4 and vitamin A plasma levels in COVID-19 patients warrant further investigation to explore potential therapeutic approaches of vitamin A supplementation during acute infection. Further prospective studies should follow to evaluate the treatment effect of vitamin A supplementation in acute COVID-19 infection.

## 5. Conclusions

In conclusion, to our knowledge, we have demonstrated for the first time that RBP4 and vitamin A levels are significantly decreased in severely ill COVID-19 patients. In this group of patients, decreased RBP4 levels could be causative for vitamin A deficiency. Substitution therapy could be another component in the treatment of COVID-19 patients.

## Figures and Tables

**Figure 1 nutrients-14-02007-f001:**
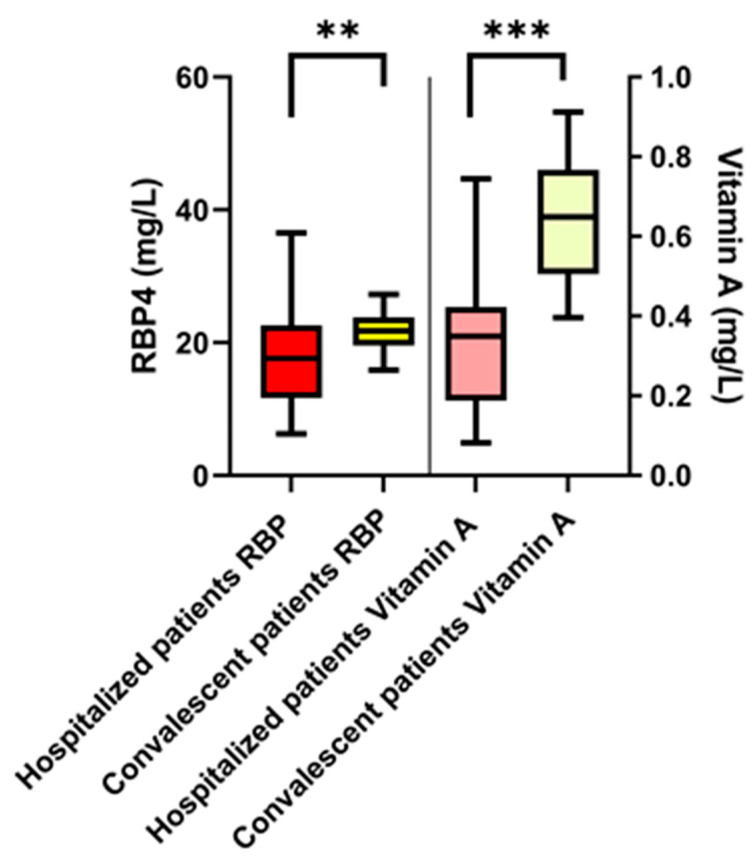
RBP4 and vitamin A plasma levels in acutely ill patients compared to convalescent patients. The Mann–Whitney U (Wilcoxon) test showed a significant difference for both RBP4 and vitamin A. RBP, retinol-binding protein 4; ** *p* < 0.01, *** *p* < 0.001.

**Figure 2 nutrients-14-02007-f002:**
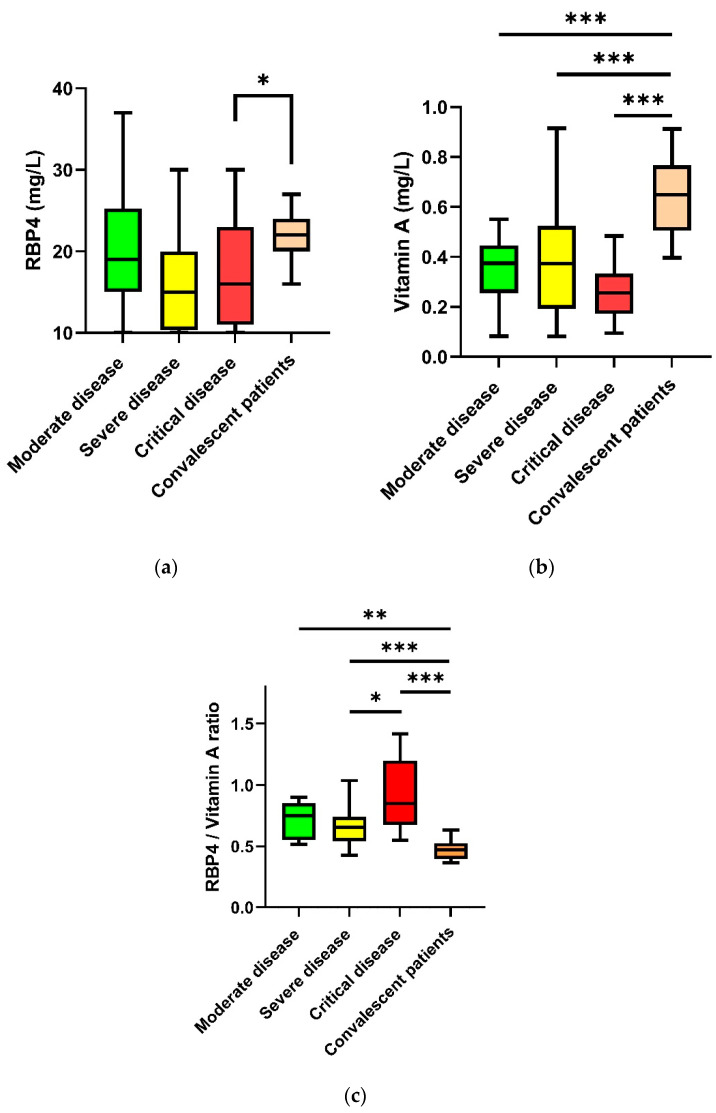
RBP4 (**a**), vitamin A (**b**) plasma levels and molar RBP4/vitamin A ratio (**c**) in hospitalized patients (moderate, severe, critical disease) and convalescent patients. Subgroups were tested for differences by the Bonferroni or Games–Howell post hoc test. The RBP4 plasma levels in critically ill patients were significantly lower compared to plasma levels in convalescent patients. The vitamin A plasma levels in all subgroups of acutely ill patients were significantly lower compared to plasma levels in convalescent patients. RBP4, retinol-binding protein 4; * *p* < 0.05, ** *p* < 0.001, *** *p* < 0.001.

**Figure 3 nutrients-14-02007-f003:**
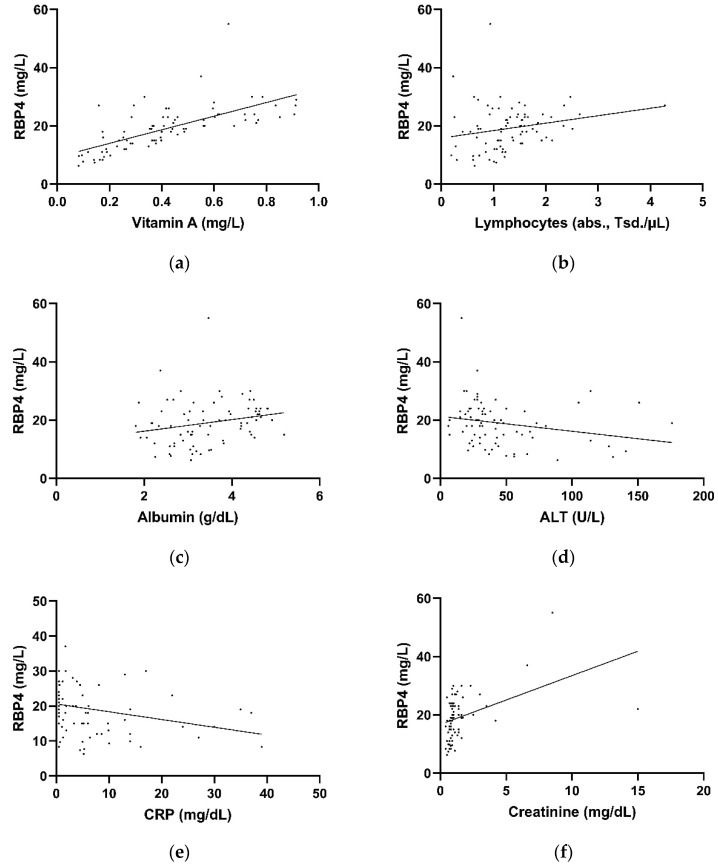
Correlation of retinol-binding protein plasma levels with vitamin A (**a**), lymphocytes (**b**), albumin (**c**), ALT (**d**), CRP (**e**), and creatinine (**f**). ALT, alanine transaminase; CRP, C-reactive protein.

**Table 1 nutrients-14-02007-t001:** Patients’ characteristics of hospitalized and convalescent COVID-19 patients.

	Moderate Disease (*n* = 20)	Severe Disease (*n* = 20)	Critical Disease (*n* = 19)	Convalescent Patients (*n* = 20)	*p*-Value
Age, years, median (min–max)	56 (21–81)	60 (44–85)	59 (41–88)	58 (50–70)	0.54
Gender, male (%)	85	70	84	90	0.38
BMI (kg/m^2^), median (IQR)	23 (21–24)	24 (22–27)	28 (25–29)	27 (25–30)	<0.001
Interval from first symptom to acquisition of blood sample, days, median (IQR)	4 (2–9)	7 (3–12)	13 (9–22)	46 (42–53)	0.003
Cardiovascular disease (abs.)	2	5	3	0	0.11
Respiratory disease (abs.)	3	6	1	0	0.03
Kidney insufficiency (abs.)	3	3	0	0	0.1
Neoplasm (abs.)	3	1	2	0	0.02
Diabetes (abs.)	2	4	2	0	0.22
Death (abs.)	0	0	6	0	<0.001
Leukocytes (×10^9^/L), median (IQR)	4.7 (2.7–6.3)	5.6 (3.7–6.0)	9.6 (7.0–11.9)	5.52 (4.9–67.2)	<0.001
Lymphocytes (rel., %), median (IQR)	25.7 (18.1–32.1)	22.05 (18.65–27.53)	9.1 (6.9–13.9)	28.5 (20.7–34.5)	<0.001
Creatinine (mg/dL), median (IQR)	0.9 (0.8–1.3)	0.4 (0.7–1.6)	0.9 (0.7–1.7)	0.9 (0.8–1)	0.99
Ferritin (µg/L), median (IQR)	380 (232–735)	682 (248–824)	956 (688–2111)	160 (106–432)	<0.001
Interleukin-6 (pg/mL), median (IQR)	15 (8–27)	31 (16–82)	95 (38–224)	2 (2–2)	<0.001
C-reactive protein (mg/dL), median (IQR)	1.3 (0.5–3.8)	5.1 (3.1–7.3)	14.2 (5.6–26.9)	0.5 (0.5–0.5)	<0.001
PCHe (U/L), median (IQR)	5302 (3928–7237)	6247 (4933–8324)	3446 (2766–4711)	8962 (7853–10329)	<0.001
ALT (U/L), median (IQR)	33 (23–64)	34 (28–45)	48 (26–70)	29 (22–33)	0.09
Albumin (g/dL), median (IQR)	3.1 (2.8–3.9)	3.5 (3.1–3.8)	2.6 (2.2–2.9)	4.5 (4.4–4.6)	< 0.001
RBP4 (mg/L)	19.38 (14.59–25.28)	15.13 (10.23–20.39)	15.5 (10.5–22.5)	21.80 (19.56–23.84)	0.02
Vitamin A (mg/L), median (IQR)	0.37 (0.26–0.45)	0.37 (0.19–0.53)	0.26 (0.17–0.33)	0.65 (0.51–0.77)	< 0.001

BMI, body mass index; IQR, interquartile range; LDH, lactate dehydrogenase; PCHe, pseudocholinesterase, ALT, alanine aminotransferase; RBP, retinol binding protein.

## Data Availability

Data cannot be made public as personal patient data are included.

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
