# Peer review of "Significantly Reduced Retinol Binding Protein 4 (RBP4) Levels in Critically Ill COVID-19 Patients"

_nutrients, 2022, doi:10.3390/nu14102007_

Round 1

Reviewer 1 Report

The authors have done a satisfactory work, in describing the role of retinoids in COVID-19. I have some major comments that will help the authors to modify the article and will be helpful for the readers.

1.In the Abstract part, I would prefer the deatils about retinoids followed by their role in covid-19 as they are helpful in modulating innate immunity against viral infections.

2. In the Introduction part the authors should include more info for ARDS and try to correlate the mechanism of retinoid receptor pathway against COVID.

3.The authors must include the receptor pathway and its binding sites for the covid-19 viral molecules.

4.Do authors observe any sexual dimorphism means is there any difference in binding in males or females, as the majority of the patients in the case study are male patients?

Author Response

Thursday, May 5, 2022

Thank you for your interest in us submitting a revised version of our manuscript, "Significantly reduced retinol binding protein 4 (RBP4) levels in critically ill COVID-19 patients". We have addressed all the reviewers’ concerns point-by-point, as indicated below. Relevant changes made in the manuscript are highlighted using red text.

Richard Vollenberg

Reviewer 2 Report

The manuscript by Vollenberg et al entitled “Significantly reduced retinol binding protein 4 (RBP4) levels in 2 critically ill COVID-19 patients” presents data of retinol binding protein 4 (RBP4) concentrations in the blood (plasma) of human individuals infected with SARS-CoV-2 virus during and after COVID-19 disease.  Based on the plasma RBP4 concentrations assessed by ELISA and plasma retinol (ROH) concentrations assessed by HPLC, the authors concluded that severe COVID-19 disease is associated with significantly lower levels of plasma RBP4 and ROH concentrations, but only in critically ill patients. In addition, plasma RBP4 concentrations detected in this study were not different in patients with moderate or severe COVID-19, however plasma ROH concentrations were significantly lower in these two groups of patients as compared to convalescent patients.  The significance of the study is new data on plasma RBP4 concentrations in human individuals exposed to SARS-CoV-2. The data presented in the current manuscript go along the lines of the commonly accepted role of RBP4 as a negative acute phase response protein. However, there are several concerns that have to be addressed.

  1. The authors use plasma RBP4 concentrations of convalescent patients (patients after mild COVID-19 disease ) as control (reference) values. However, it is indicated that the patients from the convalescence group had significantly higher BMI. It is well known that people with increased adiposity have significantly higher plasma RBP4 concentrations due to adipose-derived RBP4. Therefore comparing plasma RBP4 concentrations to concentrations in the group with potentially higher RBP4 levels may bring a confounder. This should be addressed.
  2. It is known that the plasma RBP4:ROH molar ratio is usually about 1:1. The authors should also present data as a molar ratio. It is clear that the drop in plasma ROH (Fig. 2b) is more pronounced than the drop in plasma RBP4 (Fig. 2a) suggesting that more apo-RBP4 may circulate in plasma of COVID-19 patients.
  3. There are several misleading statements in the manuscript that have to be corrected:
    • line 42, the statement “Vitamin A, which belongs to the retinyl ester family and is also known as retinoic acid (RA)” is incorrect. Vitamin A, which is retinol by definition, can be esterified in the body into retinyl esters, a major vitamin A storage form, or can be oxidized to retinoic acid, a transcriptionally active form that acts as a ligand for specific nuclear hormone receptors, known as retinoic acid receptors (RAR). In the literature, retinol, retinyl esters, and retinoic acid are collectively referred to as retinoids;
    • line 47 “free vitamin A binds to lipoproteins”, line 92 “Vitamin A (both free/unbound vitamin A and vitamin A bound to retinol binding protein [RBP])” and line 96 “free retinol”. There is no free or unbound retinol in the circulation under physiological conditions. Retinol is a lipid and therefore cannot be unbound in the hydrophilic environment. Vitamin A can be transported in the circulation either as retinol bound to RBP4 or in the form of retinyl esters as a part of lipoprotein, including chylomicrons. If the patients did not fast before the blood draw (this should be indicated in the materials and methods) then the blood could contain retinyl esters, which, in turn, could be hydrolyzed to retinol during the analytical procedures.
    • Line 241 “retinocid acid-inducible gene-1 receptors” should be corrected into “retinoic acid-inducible gene-1 receptors”.
  4. Lines 268-278 are repetitive fragment which is similar to lines 240-251

Author Response

(The authors gave the same response as above.)

Round 2

Reviewer 1 Report

The authors have now rectified all the queries. I would like to suggest its acceptance, after one small point in abstract part the authors have written syn. retinol., please try not to use abbreviations in abstract or provide it before usage.

Reviewer 2 Report

All of the comments have been properly addressed.